# Tracking Changes of Hidden Food: Spatial Pattern Learning in Two Macaw Species

Pizza Ka Yee Chow [1,2,*,†], James R. Davies [1,2,3,†], Awani Bapat [1,2,4] and Auguste M. P. von Bayern [1,2]

1   Max Planck Institute for Ornithology, Eberhard-Gwinner-Straße, 82319 Seewiesen, Germany;
    jd940@cam.ac.uk (J.R.D.); awanibapat@gmail.com (A.B.); avbayern@orn.mpg.de (A.M.P.v.B.)
2   Max-Planck Comparative Cognition Research Station, Loro Parque Fundación,
    38400 Puerto de la Cruz, Tenerife, Spain
3   Comparative Cognition Lab, Department of Psychology, University of Cambridge, Cambridge CB2 3EB, UK
4   Department of Behavioral and Cognitive Biology, University of Vienna, Althanstrasse 14, 1090 Vienna, Austria
*   Correspondence: kyc202.pc@gmail.com
†   Share first authorship.





**Simple Summary:** Efficiency in locating available food sources, which vary spatially and temporally, using spatial distribution patterns may differ depending on a species' diet and habitat specialisation. We hypothesised that more generalist species would acquire spatial information faster than more specialist species, due to being more explorative when changes occur. We tested this hypothesis by presenting a 'poke box' to relatively more generalist Great Green Macaws and relatively more specialist Blue-throated Macaws. The 'poke box' contained hidden food placed within wells that formed two patterns. The two patterns changed on a mid-week schedule. We found that (1) the two patterns varied in their difficulty; and (2) the more generalist Great Green Macaws took fewer trials to learn the easier pattern and made more mean correct responses in the difficult pattern, than the more specialist Blue-throated Macaws, thus supporting our hypothesis. The Great Green Macaws' better learning performance may be explained by more exploration and prioritising accuracy over speed. These results suggest how variation in diet and habitat specialisation may relate to a species' ability to adapt to spatial changes of food resources, which will be useful for the conservation efforts for the two critically endangered species to understand their abilities to cope with environmental change.

**Abstract:** Food availability may vary spatially and temporally within an environment. Efficiency in locating alternative food sources using spatial information (e.g., distribution patterns) may vary according to a species' diet and habitat specialisation. Hypothetically, more generalist species would learn faster than more specialist species due to being more explorative when changes occur. We tested this hypothesis in two closely related macaw species, differing in their degree of diet and habitat specialisation; the more generalist Great Green Macaw and the more specialist Blue-throated Macaw. We examined their spatial pattern learning performance under predictable temporal and spatial change, using a 'poke box' that contained hidden food placed within wells. Each week, the rewarded wells formed two patterns (A and B), which were changed on a mid-week schedule. We found that the two patterns varied in their difficulty. We also found that the more generalist Great Green Macaws took fewer trials to learn the easier pattern and made more mean correct responses in the difficult pattern than the more specialist Blue-throated Macaws, thus supporting our hypothesis. The better learning performance of the Great Green Macaws may be explained by more exploration and trading-off accuracy for speed. These results suggest how variation in diet and habitat specialisation may relate to a species' ability to adapt to spatial variation in food availability.

**Keywords:** parrots; *Ara ambiguus*; *Ara glaucogularis*; memory; foraging; generalist; specialist; comparative cognition

## 1. Introduction

Many species encounter spatial and temporal fluctuations of food availability in their natural environment. When food becomes temporarily unavailable at a location, behavioural flexibility is required to locate alternative food sources in order to survive. To deal with the variability in the environment, foragers may utilise spatial (e.g., distribution patterns) and specific (e.g., visual cues) environmental information to increase foraging efficiency [1–4]. The extent to which a species may use this information to locate food resources, may depend on ecological factors such as diet and habitat specialisation. For example, species that are more specialised in diet have been shown to pay more attention to specific information than generalists [5]. Being more sensitive to specific or relevant information may allow specialists to excel at manipulating certain food types efficiently in stable environments [6,7], or when encountering different but similar problems [8]. While more generalist species have been found to be slower at making decisions than specialists [9], they appear to be characterised by showing more exploratory behaviour [10] and explorative foraging techniques [11]. This may allow them to adapt to changes faster than more specialist species [12]. However, the role of ecological factors such as diet and habitat specialisation, in relation to a species' aptitude in acquiring environmental information when food distributions change, remains unclear.

The aim of this study is to examine the role of diet and habitat specialisation in relation to acquiring predictably changing spatial information. To do this, we adapted one of the well-established spatial pattern learning paradigms, which require individuals to search for hidden food (e.g., in matrices of poles, wells or on checkboards) that vary spatially and temporally e.g., [13–16]. We also examined how individuals acquired spatial information in such a situation (e.g., examining their behaviour in relation to exploration and speed-accuracy trade-offs).

We compared spatial learning performance in two closely related macaw species, considered to vary in their diet and habitat specialisation: Great Green Macaws (*Ara ambiguus*) (hereafter, GG) and Blue-throated Macaws (*A. glaucogularis*) (hereafter, BT). Investigation into their ability and speed in utilising environmental information to locate food sources, and in particular when a change occurs, will be useful for the conservation efforts of the two critically endangered species (GG: [17,18]; BT: [19,20]) to understand their ability in adapting to ecological change.

Both species live in fairly stable social groups and are often observed foraging in flocks (BT: [21]; GG: [22]). As documentation of the full diet habits of both species in the wild is limited, especially BT, Levin's niche breadth [23] to calculate the relative diet generalisation of each species is currently unavailable. However, available records appear to indicate that GG adapt to a wider variety of habitats, which vary in their seasonality, than BT, suggesting that GG may exploit a greater variety of foods than BT. For example, GG inhabit seasonally dry forests on the southern Pacific coast of Ecuador, as well as less seasonal wet tropical forests from Honduras to Colombia [22,24,25], whereas BT (historically and currently) only inhabit seasonal savannah habitats in relatively restricted areas of Beni, Bolivia [21]. Here, the main food source of BT, motacú palms (*Attalea phalerata*), produce fruit continuously throughout the year [26]. This suggests that the need to explore alternative food sources (and thus food variety) for BT may be lower than GG. Importantly, although BT show flexibility in utilising alternative resources during nesting [27], as well as occasionally consuming other foods such as seeds, nuts, and flowers (which GG also do) [28,29], they appear to be more specialised in diet than GG. BT consume more fruits than seeds [21,26,30,31] and not only do they rely on the presence of motacú palms to survive, but their alternative food sources also appear to be other species of palms (e.g., *Acrocomia aculeata*, and *Mauritia fleuxosa*) [21,26,30,31]. In contrast, GG feed more on seeds and nuts than fruit [22], and despite showing a preference for feeding in almendro trees (*Dipteryx oleifera*) and beach almond trees (*Terminalia catappa*) during their respective fruiting seasons, they otherwise feed on a wide variety of plant species, such as titor trees (e.g., *Sacoglottis trichogyna*), and quaruba (e.g., *Vochysia ferruginea*), to name a

few (see food list in Humedal Maquenque Anexo #2, also see [22,25,32–34]. Such variation in diet composition between the two species may also be related to differing energetic requirements in relation to their body size [28]. Nevertheless, the discussed differences in ecologies suggest that the two species are suitable for us to examine our hypothesis: that variation in diet and habitat specialisation would affect a species' speed in acquiring spatial information. Specifically, we predicted that in a test situation featuring constant and predictable recurring temporal change, the more generalist GG would be more explorative and show better spatial pattern learning performance than the more specialist BT. We additionally examined their learning strategies to understand how individuals process spatial information.

## 2. Methods

### 2.1. Study Species and Housing

Six individuals from each species (Table 1) participated in this study between August 2019 and March 2020. These macaws were hand-raised by the Loro Parque Foundation in Tenerife (Spain) and were group-housed, by species, in 5 connected semi-outdoor aviaries at the Comparative Cognition Research Station situated in Loro Parque, Puerto de la Cruz, Tenerife (Note S1). The parrots received natural light and ambient outdoor temperature at the back part of each aviary. Arcadia Zoo Bars (Arcadia 54 W Freshwater Pro and Arcadia 54 W D3 Reptile lamp) were installed at the front of the aviary to ensure the parrots received sufficient exposure to UV light. This experiment did not involve any invasive methods, negative reinforcement, or punishment. The parrots were fed twice a day with fresh fruit, seeds, and vegetables. Additional high-quality food rewards (nut pieces) could be obtained during the experiment. Water was accessible ad libitum at all times. We provided enrichment and monitored the parrots' health at least twice per week.

**Table 1.** Individual characteristics (name, age, sex, and mean weight in grams) for each species. Note that the mean weight is the average of the weights taken during the regular health checks over the study period (August 2019 to March 2020).

| Great Green Macaws (*Ara ambiguus*) | | | | Blue-Throated Macaws (*A. glaucogularis*) | | | |
|---|---|---|---|---|---|---|---|
| **Name** | **Age** | **Sex** | **Weight (g)** | **Name** | **Age** | **Sex** | **Weight (g)** |
| Hagrid | 5 | M | 1213 | Mowgli | 5 | M | 762 |
| Enya | 5 | F | 1160 | Charlie | 5 | M | 783 |
| Luna | 5 | F | 1129 | Long John | 6 | M | 746 |
| Madame | 5 | F | 1064 | Gargamel | 7 | M | 849 |
| Alba | 5 | F | 1153 | Lady | 5 | F | 728 |
| Rita | 5 | F | 1119 | Mr. Huang | 6 | M | 812 |

### 2.2. Test Room Set Up

Inside the test room (Length × Width × Height: 1.5 × 1.5 × 1.5 m), one wall was light green and the opposite wall was white. There was a sound-buffered, one-way glass panel connecting the two walls, which allowed zoo visitors to observe the experiments live (Figure 1A). Three tables (each 87 × 49 × 150 cm) were placed in the room, with the middle table being a sliding table. The experimental apparatus, i.e., the 'poke box' (see below) was attached to the centre of another white table that was placed on top of the sliding table, so that it could be moved forward to be replenished with food rewards during inter-trial intervals. The sliding table was marked by three strips of black tape: two thin strips running parallel along the outer sides of the poke box facing the walls, and a third thick strip running perpendicular to the others and underneath the poke box (Figure 1A). We used two (out of four) surveillance cameras, in the top corner and side positions of the test room, to observe and record the behaviour of the parrots. The test room had additional visual elements that consisted of electric heaters, lights, and ventilation fixtures. These provided unique spatial features relative to the poke box at the centre of the room, which

the parrots may use as 'landmarks'. The lights were flicker-free and covered the parrots' full range of visible light (Arcadia 39 W Freshwater Pro and Arcadia 39 W D3 Reptile lamp).

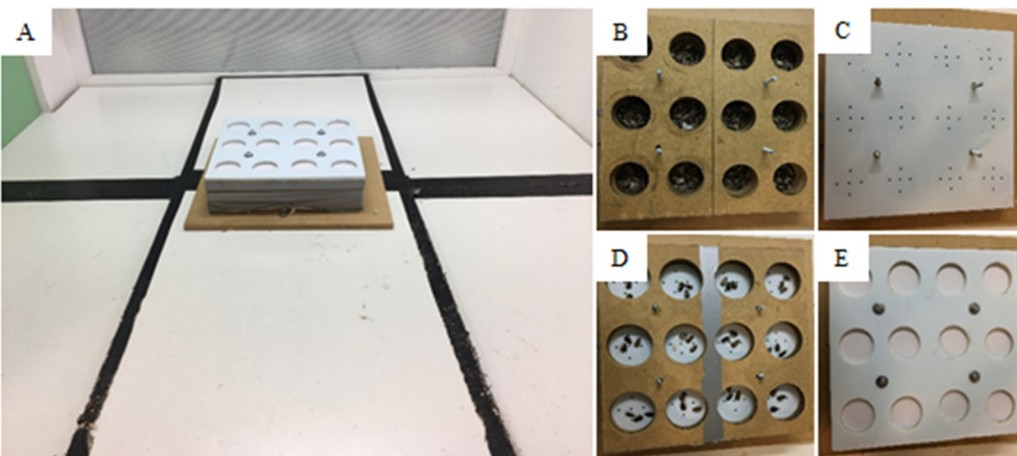

**Figure 1.** Poke box set up for testing the macaws' spatial learning performance. (**A**) The set up in the test room; (**B**) base wooden layer with 12 wells filled with inaccessible food (seeds for habituation and nut pieces for training); (**C**) the divider with holes used to separate the top and the bottom layers with holes to release scent; (**D**) the upper wooden layer containing the accessible rewards (seeds or nuts); and (**E**) the top plexiglass layer with paper covering the wells.

### 2.3. Experimental Apparatus

To examine the parrots' spatial learning performance, the 'poke box' design was adapted from [35,36]. It was a flat square box (30.5 × 30 × 3.7 cm; Figure 1A) containing 12 wells (each 6 cm diameter) spaced equally in a 3 × 4 manner. The box (from bottom to top) consisted of four major parts: (1) the bottom consisted of two wooden layers (each 30.5 × 30 × 1 cm) (hereafter, the bottom). Each well in this bottom part was filled with chopped-up nuts which were inaccessible (Figure 1B) but released a scent. (2) A white Plexiglas sheet (0.2 mm thick) that had 12 sets of five holes (each 0.2 mm diameter) in clusters corresponding to the wells in the bottom; this allowed the scent of the nuts to pass through to the top wells (Figure 1C), thus serving as a control for the use of olfactory cues in this task. (3) The top part consisted of another wooden layer (30.5 × 30 × 1 cm) onto which a white Plexiglas sheet (0.5 mm thick) was placed (hereafter, the top). The wells in this part contained accessible rewards (Figure 1D). (4) White paper was placed under the top Plexiglas sheet to cover the rewards (Figure 1E), and thus blocking visual access to the rewards in the wells (Figure 1E). The whole box was then placed in the middle of the table in the test room, secured with metal screws and brackets on each side attached to a wooden board (40 × 40 × 0.8 cm), fixed to the table.

### 2.4. Procedures

Each bird was tested individually and had been previously trained to voluntarily enter a transporting cage (1.5 × 1 × 1.5 m; Figure 2A). This transporting cage had a sliding board on one side that could be pulled up to separate a parrot from the main aviary and move it to the test room. The cage was then parked next to a wooden window integrated into the door of the test room (0.5 × 0.5 m; Figure 2A). Once the sliding board was lowered, the parrot entered the test room via a wooden ladder (Figure 2B).

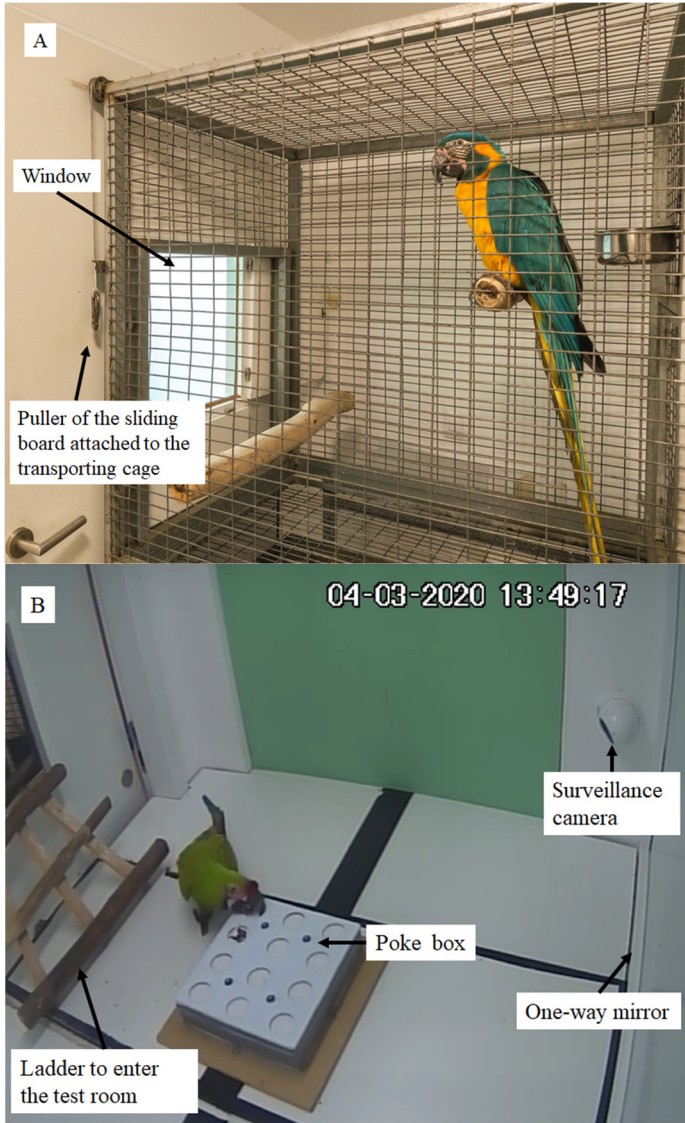

**Figure 2.** (**A**) Mowgli, a Blue-throated Macaw, in the transporting cage parked next to the door of the test room, with the sliding board of the transporting cage lowered down. The door has a window adjacent to the cage, in which the macaw can enter the test room. (**B**) A screenshot taken from the test room surveillance camera, showing Madame, a Great Green Macaw, completing a trial.

The parrots were first habituated to the box and trained to obtain food (seeds) from the wells by tearing paper (Note S2). The main training started once a parrot tore the paper, using its beak, of 10 (out of 12) or more wells for two or more consecutive days. As the parrots appeared to struggle to tear dry paper, we applied a ring of water over each well in a standardised manner to facilitate tearing (but ensured that tearing the paper still required effort), after the box was set up (Figure 1E). In the training phase, two spatial patterns of rewarded wells (A and B; Figure 3) were created. Pattern A consisted of the two columns at the outer edges of the box (6 wells), whereas pattern B consisted of the two middle columns of the box (6 wells). The parrots experienced three trials of each pattern for three consecutive days (see below for further information). The pattern was then switched for the next 3 days of the week, with no testing on the last day of the week. We counterbalanced the pattern that individuals experienced first, within each species. The order of presenting the patterns was fixed for each individual across the training weeks.

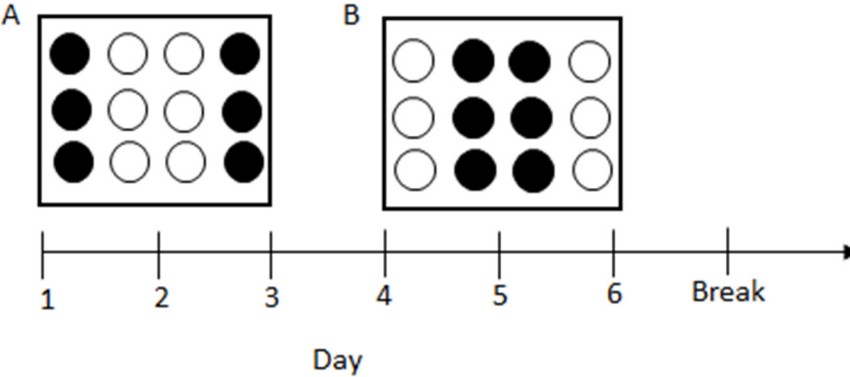

**Figure 3.** The two spatial patterns (**A**,**B**) containing rewarded wells (represented in black) during the main training. The pattern (**A**) or (**B**) that an individual experienced first was counterbalanced within each species (i.e., three parrots received pattern A first, whereas the other three parrots received pattern B first). The order of presenting the patterns was fixed for each individual across the training weeks. Each week, each parrot experienced each pattern for 3 consecutive days (3 trials per day) with a daybreak.

Before each trial, we baited the wells according to the pattern (A or B) that the individual received on that day. We cleaned the top of the box and the tables before each trial started; this minimised additional cues being provided to the parrots during training. We lowered the board of the transporting cage to initiate the start of each trial. The trial ended with the parrot voluntarily re-entering the transporting cage. Between trials, we blocked visual access to the test box by closing the window of the test room; this prevented the parrots from directly observing the locations of the rewards. Each inter-trial interval did not last longer than 2 min during training trials. Once the parrot entered the test room, the experimenter turned away (to avoid unconscious cueing towards the food locations) and recorded its choices while directly observing via a smartphone connected to the surveillance cameras (Figure 1B). We repeated this procedure 3 times for each session. If a parrot reached $\rho \geq 80\%$ (see below) on the second and third trial, we ran a fourth trial so as to determine whether the individual reached the criterion ($\rho \geq 80\%$ for 3 consecutive trials). We continued the experiment for 10 weeks, regardless of whether the parrots had reached the criterion, to compare learning performance with increased experience. A third rater (AB) re-coded the order of choices from the available videos (2.5% of the videos were lost due to system errors and unforeseen circumstances e.g., vet visits) and obtained high inter-rater reliability (ICC: BT = 0.999, GG = 0.999).

Behavioural Measurements

*Learning performance.* We considered a parrot to have learned a pattern if it reached rho ($\rho$) $\geq 0.8$ (or 80%) correct (rewarded) responses across three consecutive trials. This $\rho$ value is positively related to Mann–Whitney U test, with an emphasis on consecutive correct responses being made at the beginning of a trial (Note S3). The initial two choices of each trial were critical in determining whether a parrot would reach the criterion. In a nutshell, $\rho \geq 80\%$ only allows a parrot to make one incorrect choice at the beginning of a trial. As this criterion may be too stringent (see results below), we also assessed the parrots' learning performance by considering two other adjusted learning criteria, with $\rho \geq 0.75$ (or 75%) and $\rho \geq 0.7$ (or 70%). For each criterion, we counted the number of trials that a bird took to reach the learning criterion for each pattern. For each individual, the mean $\rho$ value of each session was used to assess their learning performance with increased experience for each pattern.

*Exploration.* To examine the macaws' explorative behaviours, we recorded two behaviours after each of them had completed the trial (as additional exploration after task completion may be useful to locate rewards): (1) the mean time (seconds) from when the

parrot opened the last well to the time it left the box when they experienced a change on the first week of training (i.e., the mean time of two changes: a change of seeds being available in all wells during the habituation phase, to nuts being available in half of the wells during the main training, and a change from food locations in pattern A to food locations in pattern B), as 'exploration after a change'; and (2) the mean time (in seconds) they spent on the table after completing the task across all trials.

*Learning strategy.* To examine the macaws' learning strategy in this task, we used the search order in which a parrot opened the wells, to reflect on their movement (forward or sideways) (Figure 4). As the maximum number of correct responses was 6 in each trial, our focus was on the order (and corresponding correct or incorrect responses in each pattern) of these initial 6 choices. To obtain the mean correct responses made in the first six choices, we divided the total number of correct responses made in the first 6 choices by the total number of trials an individual participated in. We also obtained the total number of wells that a parrot opened for each trial and divided this number by its time spent in completing that trial (i.e., the duration in seconds between opening the first well and the last well), to obtain the mean time taken to open a well.

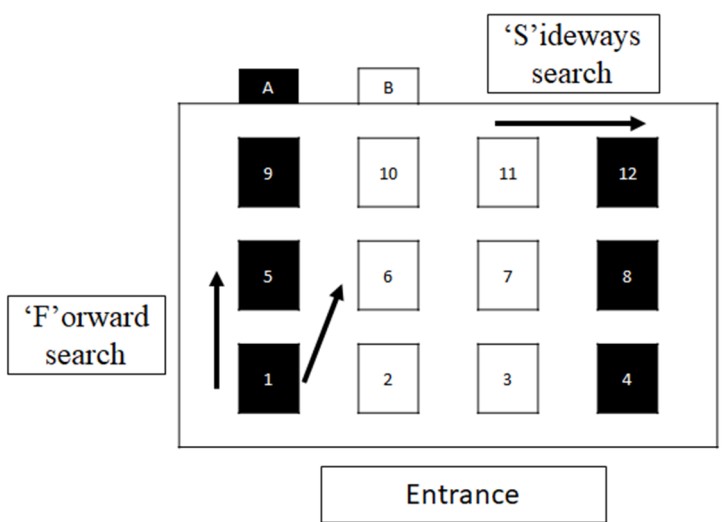

**Figure 4.** Well numbers assigned on the poke box. A parrot's movement on the poke box can be categorised into two types: moving forward (e.g., $1 \rightarrow 5$ or 6) or moving sideways (e.g., $11 \rightarrow 12$). Depending on the pattern (A or B), the search order also provides correct and incorrect responses as well as the total number of wells opened for each trial.

*2.5. Data Analyses*

We analysed all data using R studio (version 1.1.463) [37] and SPSS (version 25) (IBM Corp.). As differences in diet composition have been shown to relate to body size (and thus weight) in Neotropical parrots [28], we attempted to control for body weight in all models, either as an additional random effect in the GLMM model or as a fixed effect in the GLM models (see below). However, a Pearson's r correlation, that was conducted prior to model fitting, showed a close-to-perfect positive relationship between species (GG or BT) and weight (r = 0.97). The attempted inclusion of weight also resulted in convergence issues. Accordingly, our subsequent analyses placed the focus on our main independent variable, species, when examining the parrots' learning performance, their exploration behaviour, and learning strategy. The normality of the data was checked using the Shapiro–Wilk test when selecting the family and link distribution. To examine the parrots' learning performance across sessions, we used a Generalised Linear Mixed Model with Binomial log link distribution to explore three main fixed effects, species (GG or BT), pattern (A or B) and session number (1–30), on the mean $\rho$ value for each session across sessions of the unadjusted criterion ($\rho \geq 0.8$). Individual identity was set as a random variable. Three

Generalised Linear Models (GLM) with Poisson log link distribution were carried out to examine the effect of species on the number of trials taken to reach each of the learning criteria ($\rho \geq 0.8$, $\geq 0.75$, or $\geq 0.7$) for each pattern (A or B). To examine species differences in exploration behaviour and learning strategy, another five GLM models with gaussian distributions were used to examine the mean time (seconds) to open a well, mean time to complete a trial, exploration after a change (of pattern), mean time (seconds) spent on the table, and the mean correct responses made in the first six choices. A two-tailed significant test was $p \leq 0.05$.

Further analyses on the parrots' learning strategy in relation to a failure of learning a pattern (see below for detail) were conducted using each bird's search order. These analyses were mostly descriptive and individual-based. We report (1) the frequency of using 'forward' or 'sideways' search; (2) sideways responses that led to incorrect responses; (3) mean consecutive correct responses made on the first six choices; and (4) frequency of consecutive correct responses made in the first six choices.

## 3. Results

### 3.1. Learning Performance

With $\rho \geq 0.8$, none of the BT and one GG (Hagrid) learned both patterns. Learning performance across training sessions did not differ by species (GLMM: Z = −0.41, $p = 0.680$), but did differ by pattern (Z = 15.27, $p < 0.001$) and training session (Z = 3.69, $p < 0.001$) (Table S1). Performance was better in pattern B than pattern A (Figure 5), and as training progressed. None of the macaws except one GG, Hagrid, learned pattern A with both adjusted learning criteria ($\rho \geq 0.75$ and $\rho \geq 0.7$). For pattern B, 4 BT and all GG reached $\rho \geq 0.75$, and all macaws reached $\rho \geq 0.7$ (see Supplementary Videos for pattern A and pattern B). As most individuals only learned pattern B, we compared the number of trials taken to reach each of the criteria between the two species. We found that the GG were significantly faster than the BT in learning pattern B regardless of whether the criterion was adjusted (GLM: $\rho \geq 0.8$: $\chi^2_1 = 60.21$, $p < 0.001$; $\rho \geq 0.75$: $\chi^2_1 = 24.18$, $p < 0.001$; $\rho \geq 0.7$: $\chi^2_1 = 7.46$, $p = 0.006$; Figure 6, Table S2).

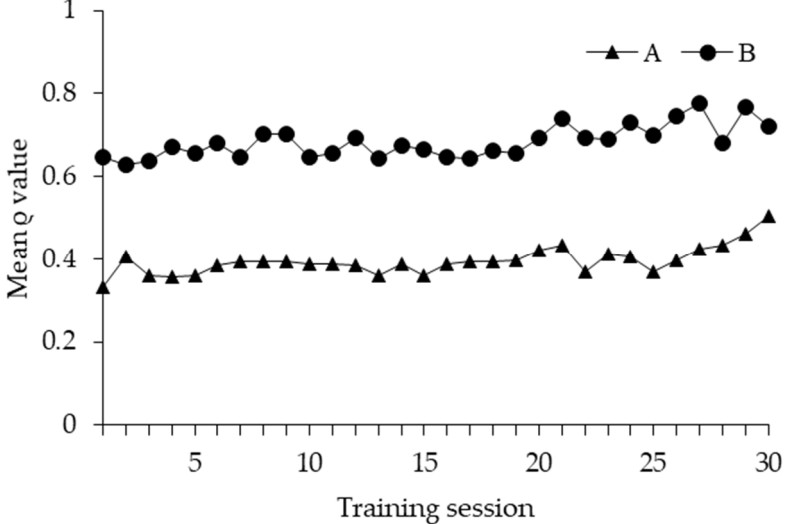

**Figure 5.** Learning performance, indicated as mean $\rho$ values of both species (N = 12), across training sessions for each pattern.

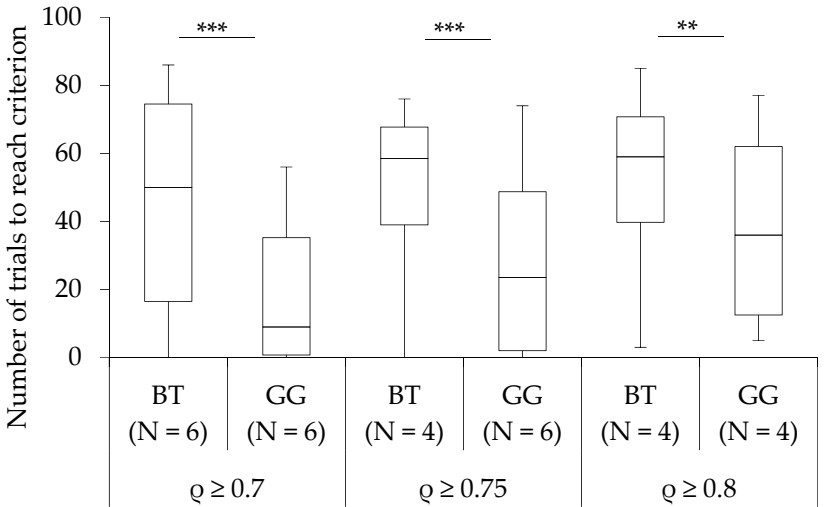

**Figure 6.** Box plot of three passing criteria, ρ ≥ 0.7 (or 70%), ρ ≥ 0.75 (or 75%), and ρ ≥ 0.8 (or 80%), to assess learning performance for pattern B for each species (BT = Blue-throated Macaws, GG = Great Green Macaws). *** *p* < 0.001, ** *p* < 0.01.

### 3.2. Learning Strategy and Exploratory Behaviour

Across all trials (of both patterns), the BT took significantly less time to open a well than the GG ($\chi^2_1$ = 9.77, *p* = 0.002) (Figure 7A, Note S4, Table S3A) and to complete each trial ($\chi^2_1$ = 10.60, *p* = 0.001, Median BT = 46.7 s, GG = 61.3 s, Figure 7B, Table S3B). When they first experienced a change (of pattern) in the first week, the mean exploration time after a change of pattern was significantly less in the BT than in the GG ($\chi^2_1$ = 4.35, *p* = 0.037, Table S3C) (Figure 8A). On average, the mean time spent on the table was also significantly less in the BT than in the GG ($\chi^2_1$ = 7.72, *p* = 0.005, Table S3D) (Figure 8B). Compared with the BT, the GG made significantly more mean correct responses in the first six responses per trial in pattern A ($\chi^2_1$ = 5.17, *p* = 0.023, Table S3E) but not in pattern B ($\chi^2_1$ = 0.95, *p* = 0.329, Table S3F) (Figure 9).

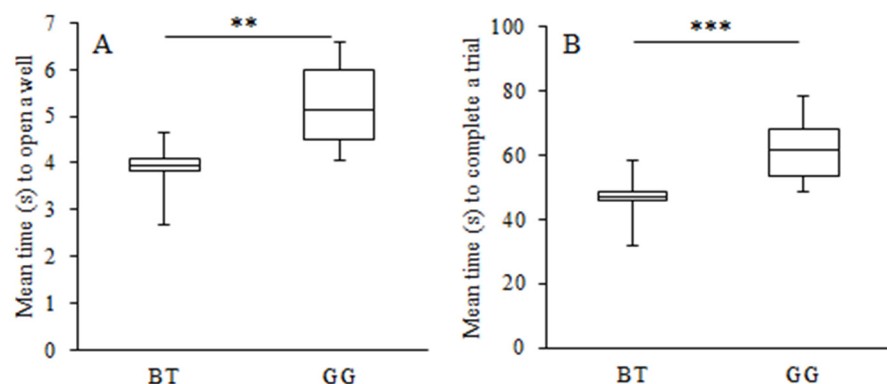

**Figure 7.** Box plot of mean time (s) to (**A**) open a well and (**B**) complete a trial (BT = Blue-throated Macaws, GG = Great Green Macaws). ** *p* < 0.01, *** *p* = 0.001.

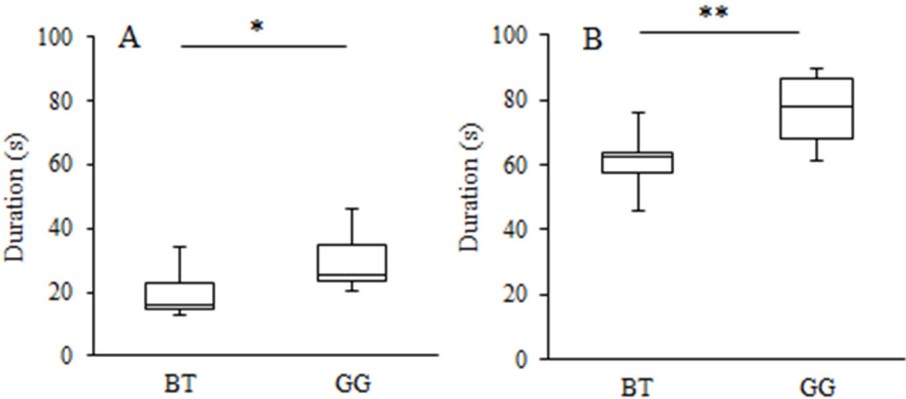

**Figure 8.** Box plots of exploration-related behaviours: (**A**) the two species' 'exploration after a change of pattern occurs, measured as the mean duration in seconds from opening the last well to leaving the box when the parrots first experience a change in the first week. (**B**) The mean time in seconds spent on the table across trials (BT = Blue-throated Macaws, GG = Great Green Macaws). * $p < 0.05$, ** $p < 0.01$.

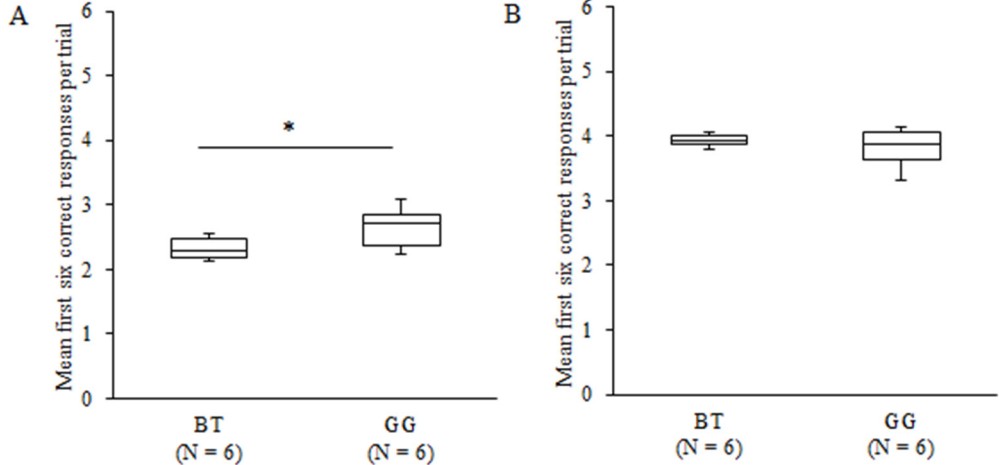

**Figure 9.** Search order analysis on the mean of the first six correct responses per trial for pattern A (**A**) and pattern B (**B**). * $p < 0.05$.

### 3.3. Search Order in Pattern A

Most macaws did not reach the criterion for pattern A, and so we analysed their search order to examine why they may have failed to do so. During a trial, the macaws frequently chose an adjacent well after making a choice (e.g., 1 → 2, 2 → 6 Figure 4). Their movement to an adjacent well could be categorised as two types: forward (e.g., 2 → 6) or sideways (e.g., 1 → 2) (Figure 4). However, sideways movements would not facilitate the macaws to reach the criterion of pattern A. We focused on the unsuccessful individuals' initial two choices of each trial, which determined whether they would reach $\rho \geq 0.8$ for that trial. Indeed, all the BT and 4 GG (except Enya) mostly showed sideways movement (median: BT: 94.3%, GG: 95.6%) (Figure 10A). Regardless of the outcome (correct or incorrect) of the first response, moving sideways to an (incorrect) adjacent well (e.g., 1 → 2, Figure 4) was predominant in all the BT (median: 87.1%) and one GG (Rita, 100%) (Figure 10B).

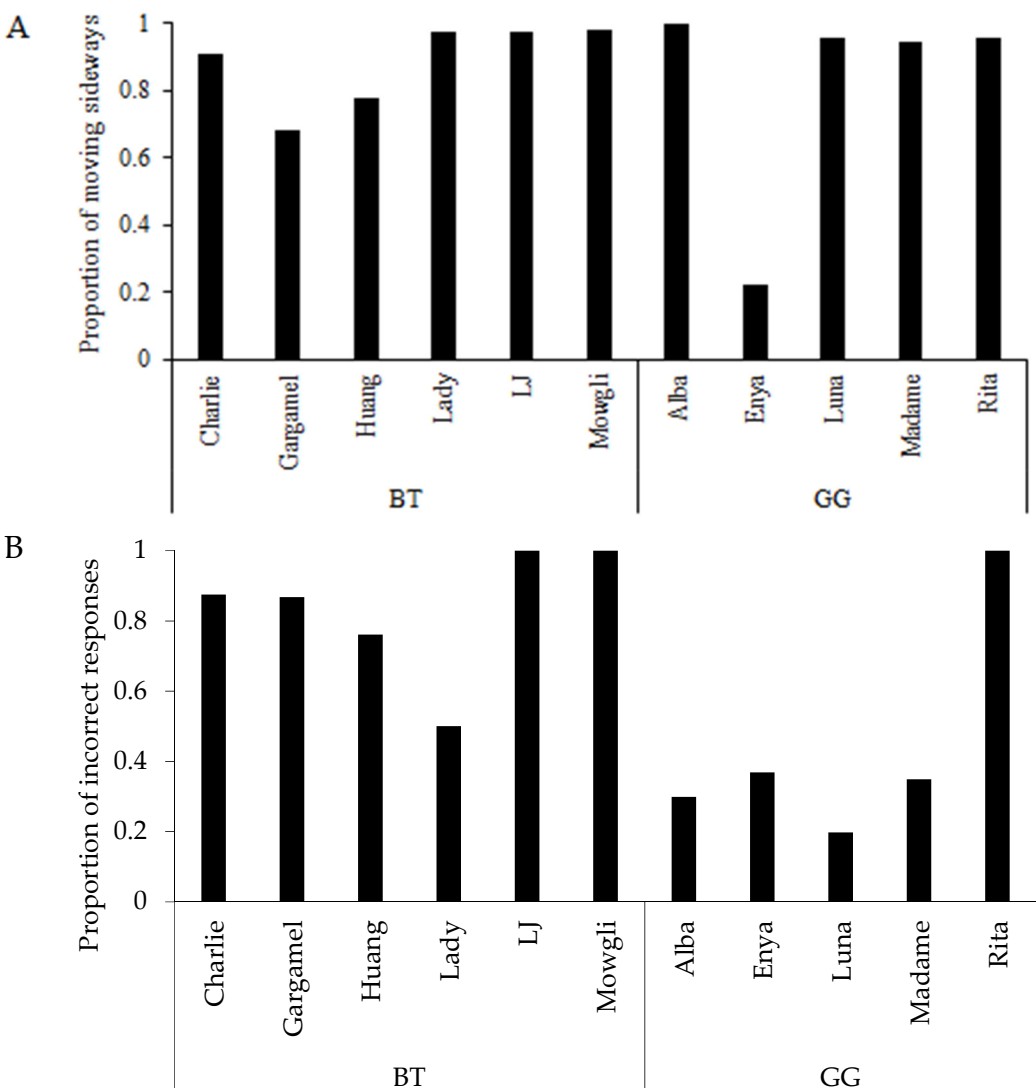

**Figure 10.** *Cont.*

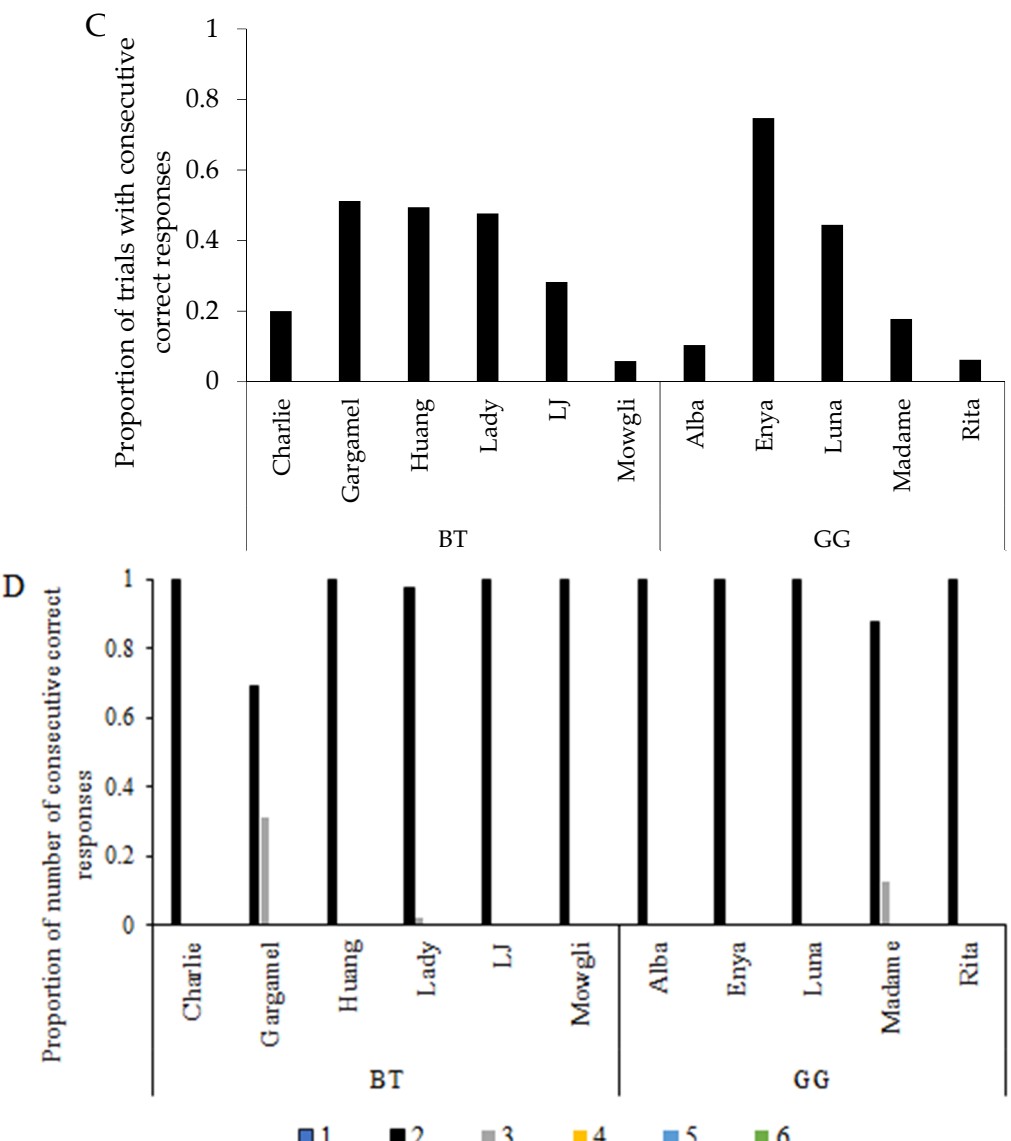

**Figure 10.** Search order in pattern A. A parrot may open another well by either going forward or sideways (see Figure 4). The proportion of (**A**) sideways movement (from the first to the second choice), (**B**) sideways responses that lead to incorrect responses; (**C**) trials with consecutive correct responses made on the first six responses; and (**D**) the consecutive number of correct responses made in the first six choices (ranged from 1–6, coded in different colour).

Perhaps the unsuccessful macaws were using an alternative strategy, i.e., opening a well to indicate which pattern they should follow, or that they may have perceived pattern A as two separated columns (Figure 4). In both cases, the search order analysis in the first six responses of each trial should show that the parrots made three or more consecutive responses. We conducted an additional analysis of the first six responses (Note S5), which showed that the macaws did make consecutive correct responses (median proportion of trials with consecutive correct responses BT: 38.0%, GG: 17.8%, Figure 10C), however, they predominantly made 2 (instead of 3 or more) consecutive correct responses (Figure 10D).

## 4. Discussion

This study provides evidence in support of the hypothesis that variation in diet and habitat specialisation affects a species' propensity to learn spatial information under predictable spatial and temporal change. Two closely related macaw species that vary in diet and habitat specialisation improved their learning performance over the course of

training, despite the two patterns varying in their difficulty. While both macaw species acquired the easier pattern (pattern B), the relatively more generalist Great Green Macaws performed better overall than the more specialist Blue-throated Macaws. The Great Green Macaws were faster in acquiring the spatial pattern B and reached a higher accuracy than the Blue-throated Macaws in pattern A.

The fact that the more generalist Great Green Macaws showed better performance in acquiring spatial information than the more specialist Blue-throated Macaws, can be partially explained by the Great Green Macaws showing more exploration in general, and in particular, after a change occurs. Generalist species are predicted to be more explorative than specialists [10]. Concerning our tested species, however, we know little about their exploration behaviour. In a previous study on flexible and novel food-extraction problem solving, the two species did not differ in most exploration measures that were directly related to the problem (e.g., duration in touching the apparatus) [38]. However, exploratory behaviours allow individuals to obtain additional information in their environment [39], which could be seen in a more indirect way. In this study, our exploration measures were recorded after the individuals completed the task. The more generalist Great Green Macaws, on average, spent more time on the table and in particular, exploring more after a change of pattern occurred, than the Blue-throated Macaws. Increased exploratory behaviours in the Great Green Macaws may have helped them to take fewer trials to learn the easier pattern B and showed more correct responses for the more difficult pattern A, than the Blue-throated Macaws. Exploratory behaviours have been proposed as a pre-adaptive trait, along with behavioural flexibility, that can lead to successful establishment in new environments (e.g., [40]). A greater tendency to explore may thus relate to the wider diet spectrum and habitat range reported for Great Green Macaws compared to Blue-throated Macaws [22,24,25]. However, the relative role of exploration in relation to diet spectrum and habitat range, alongside other behavioural and personality traits such as neophobia [41,42], or tolerance in dealing with greater climate variation and environmental uncertainty associated with relatively larger brains [43], remains to be investigated. In addition to this, although exploratory behaviours have mostly been related to a species' ecological background (e.g., diet and/or habitat specialisation) [10,39,44,45], exploratory behaviours have also been shown to vary within a species [46]. Such inter-individual variation in exploratory behaviour may relate to differences in how individuals seek and acquire information [46], which may relate to variations in making correct/incorrect choices or consecutive correct responses, as well as affecting individual specialisation in utilising different resources [47,48].

Another explanation for the species difference in performance may be related to the use of opposing speed vs. accuracy trade-off strategies, that in turn relate to the species' diet and habitat specialisation. Compared with generalists, specialists tend to make faster decisions [9], and seem to pay more attention to environmental cues during foraging [5]. The Great Green Macaws on average spent more time opening wells and took longer to complete a trial than the Blue-throated Macaws, possibly reflecting that the Great Green Macaws traded-off speed for accuracy, whereas the Blue-throated Macaws traded-off accuracy for speed. Both strategies appear to be equally adaptive as it leads to the same outcome (collecting all rewards). The fast-but-inaccurate strategy that the Blue-throated Macaws employed may reflect their readiness in deploying certain specialised foraging skills (i.e., tearing the paper of the wells in this task), which in turn allows them to afford the costs of making errors. Similar to many humans [49] and non-human animals such as bumblebees (*Bombus terrestris*) [50,51], Indian Mynas (*Sturnus tristis*) [52], Carib Grackles (*Quiscalus lugubris*) [53], Ring-tailed Coatis (*Nasua nasua*) [54], and even in basal eukaryotic organisms like slime moulds (*Physarum polycephalum*) [55], the use of such strategies varies within species. While we observe individual variation in the use of these strategies within each species, the factors that affect an individual's use of which learning strategy is not the main focus of this study and requires future investigations.

In this study, we have adapted established paradigms that are used to examine spatial pattern learning (e.g., [13–16]). While paradigms used across studies vary depending on the examined research question (and thus, comparability of performance should be cautious), the ability to show spatial pattern learning has also been shown in bumblebees [56], Sprague-Dawley rats [13,14], Rufous Hummingbirds [16], to name a few. In this study, the macaws experienced two, arguably simple, recuring patterns in a predictable manner. However, only one Great Green Macaw learned both patterns. In general, the macaws showed a better learning performance for pattern B (wells located in the two middle columns) than pattern A (wells located at the two vertical edges of the poke box). This result seems counterintuitive in that pattern A should have provided visual cues, or direct internal (i.e., box edges) and external (e.g., wall colour) references for the macaws to locate the food. The use of visual cues is expected to enhance spatial pattern learning, as has been shown in, for example, human participants showing better performance when a visual cue is associated with either rewarded or non-rewarded locations [57]. While a previous study investigating spatial pattern learning in rats has also shown that visual cues facilitated rats to not revisit locations where they had retrieved rewards, visual cues did not necessarily enhance their spatial pattern learning [58]. The fact that the unsuccessful macaws showed a low proportion of correct responses and rarely made consecutive correct responses, indicates that they were neither using a strategy such as opening a well to infer the pattern as has been shown in rats [15] nor perceiving pattern A as two separated columns. Their poor performance in learning pattern A could be explained by the macaws mostly moving sideways. Such perseverance in moving sideways, in particular in the case of the Blue-throated Macaws, may reflect their restricted inhibitory control or inflexibility that has been reported in novel food-extraction problem-solving tasks [59]. However, we also consider the possibility that the macaws did not have to inhibit themselves, as their search strategies described above ultimately allow them to yield the maximum gain. Poorer performance in pattern A may also partly relate to the distance between rewarded choices: the two rewarded choices are adjacent to each other in pattern B but spread apart in pattern A. Accordingly, the parrots had a higher chance of making correct choices in pattern B by simply employing their "moving sideways" strategy. However, this led to a much lower success in pattern A.

## 5. Conclusions

In summary, our study investigated the ability and speed of two closely related macaw species, with different ecological backgrounds, in acquiring spatial information in an ecologically relevant foraging task. The more generalist Great Green Macaws outperformed the more specialist Blue-throated Macaws in learning performance over both spatial patterns, providing support for the hypothesis that generalist species learn spatial patterns faster, and thus, may adapt to change more easily. Such knowledge may provide relevant information for the conservation of these two critically endangered species, in that Blue-throated Macaws may be more vulnerable to changes in their natural environment.

**Supplementary Materials:** The following are available online at https://www.mdpi.com/article/10.3390/birds2030021/s1, Note S1: Housing, Note S2: Habituation (Pre-training phase), Note S3: Learning performance, Note S4: Search time analyses, Note S5: Search order analyses, Table S1: Result of a GLMM model with the effect of species (GG or BT), pattern (A or B), and training session on learning performance across training sessions, Table S2: Pattern B: Results of GLM models with a Poisson distribution that includes the effect of species (GG or BT) on the three learning criteria ($\rho \geq 0.8$, $\rho \geq 0.75$, and $\rho \geq 0.7$), Table S3: Results of the GLM models with a Gaussian distribution that includes the effect of species (GG or BT) on the mean time (seconds) to open a well (A), to complete a trial (B), to explore after a change of pattern on the first week (C), spent on the table (D), as well as the mean correct responses made in the first six choices for pattern A (E) and pattern B (F), Video S1: Pattern A, Video S2: Pattern B.

**Author Contributions:** P.K.Y.C. designed the experiment with input from J.R.D. and A.M.P.v.B.; P.K.Y.C. and J.R.D. conducted the experiment, wrote the protocol, ran the analyses, and wrote the

first draft.; P.K.Y.C. and J.R.D. analysed all videos and A.B. conducted inter-rater reliability tests. All authors revised the manuscript and approved the submission. All authors have read and agreed to the published version of the manuscript.

**Funding:** This research was funded by Animal Mind Project to P.K.Y.C.

**Institutional Review Board Statement:** Following the German Animal Welfare Act of 25 May 1998, Section 5, Article 7, and the Spanish Animal Welfare Act 32/2007 of 7 November 2007, Preliminary Title, Article 3, this study was classified as a non-animal experiment, and thus it did not require approval from a relevant body. The care and use of animals were strictly adhered to the ethical guidelines of the Association for the Study of Animal Behaviour and Animal Behaviour Society.

**Data Availability Statement:** Data is available in OSF: DOI 10.17605/OSF.IO/FK62T.

**Acknowledgments:** We thank Loro Parque, Wolfgang Kiessling, and his staff for their generous support and for providing us with access to the birds and the research facilities. We thank the Loro Parque Fundación and Christoph Kiessling, for their collaboration and the staff of the Loro Parque Fundación for their support. We are grateful to T. Boehly for assisting with the logistics for the experiment, M. Petelle for building parts of the apparatus, and A. Krasheninnikova for general advice and feedback on lab logistics.

**Conflicts of Interest:** All authors declare there is no conflict of interest.

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
