# Peer review of "Tracking Changes of Hidden Food: Spatial Pattern Learning in Two Macaw Species"

_2673-6004, doi:10.3390/birds2030021_

Round 1
Reviewer 1 Report
Dear authors:
I find the idea of this paper very novel and of great interest. The experiment carried out in the article is very well thought out and meticulously done. The authors have done an extraordinary job in this regard. However, there are certain assumptions made by the authors that I think should be reviewed, since based on the existing literature I cannot agree with the authors on some points. On the other hand, I believe that some questions should be incorporated to enrich the discussion and add more information regarding the models made. My comments are in the document attached, I hope they will help the authors with these points.

Author Response
Dear Reviewer 1,
Thank you for the detailed and valuable comments on our manuscript. We have taken each comment into account when revising the manuscript. Each comment is listed in a table, which you can find it in the attached document.
Thank you for your time in reviewing our manuscript in advance.
Best wishes,
Pizza

Reviewer 2 Report
It was a pleasure to read this manuscript. In general, I found the study topic quite interesting, the implementation of the experiment and analysis methods is sound and the manuscript is well written.
I have identified only a single point that I believe could improve the presentation of the stu. It is related to the predictions presented at the end of the introduction. It seems to me that, from the Introduction up to the Results sections, the predictions are too loosely connected to (1) the underlying hypotheses and (2) the specific behavioral measurements taken in the experiment and the subsequent data analysis. Thus, I would like to see at the end of the Introduction a clear statement of the hypotheses underlying the predictions. I see that the hypotheses are hinted at in the first paragraph, but I think readers will benefit from an objective statement connecting them to the predictions. Also, in the Data analysis part of the Methods section, it would be very useful to objectively relate the behavioral measurements and factors of interest to the actual predictions presented earlier. This will help readers make better sense of how the experiment/analysis addresses the research question.
Author Response
Dear Reviewer 2,
We are grateful for your positive feedback on our manuscript. We have now taken your comments to improve the manuscript. Please kindly find our responses in the attached file.
Thank you for your time in reviewing our paper again.
Warmest wishes,
Pizza

Round 2
Reviewer 1 Report
The authors have done an excellent job addressing all the issues raised in the review, significantly improving the manuscript. They have satisfactorily incorporated some of the points specified in both the introduction and discussion, as well as included more information in the results. I am also grateful for the clarifications regarding some of the questions or issues raised. I hope to see it published soon in this journal.
Author Response
Dear Reviewer 1,
On behalf of the co-authors, I thank for your positive feedback on our manuscript. We are grateful for your comments which have significantly improved the manuscript.
Best wishes,
Pizza